# Reducing the Risk of Benthic Algae Outbreaks by Regulating the Flow Velocity in a Simulated South–North Water Diversion Open Channel

**DOI:** 10.3390/ijerph20043564

**Published:** 2023-02-17

**Authors:** Longfei Sun, Leixiang Wu, Xiaobo Liu, Wei Huang, Dayu Zhu, Zhuowei Wang, Ronghao Guan, Xingchen Liu

**Affiliations:** 1Department of Water Ecology and Environment, China Institute of Water Resources and Hydropower Research, Beijing 100038, China; 2State Key Laboratory of Simulation and Regulation of Water Cycle in River Basin, China Institute of Water Resources and Hydropower Research, Beijing 100038, China

**Keywords:** flow velocity, benthic algae, South–North water diversion, environmental factors, river ecosystem

## Abstract

The reduction in open-channel flow velocity due to China’s South-to-North Water Diversion Project (SNP) increases the risk of benthic algal community blooms resulting in drinking water safety issues. Consequently, it has attracted attention from all walks of life. However, regulatory measures to mitigate the risk of algal blooms and the main risk-causing factors are unclear. This study simulated the river ecosystem of the SNP channel through water diversion. Simulated gradient-increasing river flow velocity affects environmental factors and benthic algal alterations, and can be used to explore the feasibility of regulating the flow velocity to reduce the risk of algal blooms. We found that the algal biomasses in the velocity environments of 0.211 and 0.418 m/s decreased by 30.19% and 39.88%, respectively. Community structure alterations from diatoms to filamentous green algae were 75.56% and 87.53%, respectively. We observed significant differences in biodiversity, especially in terms of richness and evenness. The α diversity index of a species is influenced by physical and chemical environmental factors (especially flow velocity). Our study revealed that flow velocity is the main factor affecting the growth and outbreak of benthic algae. The risk of algal blooms in open channels can be effectively mitigated by regulating the flow velocity. This provides a theoretical basis for ensuring the water safety of large-scale water conservancy projects.

## 1. Introduction

Rivers are vital natural resources that provide important ecosystem services to humans, including fish, drinking water, flood control, and various economic activities [1,2,3]. However, because of human activities, such as the construction of China’s Three Gorges Dam and the “South-to-North Water Diversion” project (SNP), river ecosystems have been continuously degraded. These activities reduce the flow velocity of native rivers and introduce excess nutrients and hydrological regulation, resulting in many ecological problems, especially benthic algae and cyanobacterial blooms, threatening the water supply function and balance of the river ecology [4,5]. Despite the significance of rivers and the many threats they face, we still lack a comprehensive and detailed understanding of the river’s ecological characteristics and ecosystem successional dynamics after the effects of human economic construction [6,7]. Determining and predicting the response of river ecosystems to environmental changes is necessary to understand the ecosystem dynamics and establish effective means of future human construction.

The dam construction and SNP decreased river flow velocity, increasing the risk of benthic algal blooms attached to the bottom layer. In the SNP, the transferred water serves as drinking water for residents. However, the declining flow velocity causes benthic algae outbreaks in long-distance open channels, raising concerns about the safety of drinking water. “Epilithic biofilm” is an umbrella term for biomes growing on gravels, pebbles, and rocks in shallow riverbeds, and includes algae, bacteria, and micro-animals. This community serves as a source of primary productivity of the river [8,9] and a food source for fish and benthic invertebrates [10], with benthic algae being the main component. Benthic algae can respond quickly when affected by environmental factors, such as light, nutrients, and turbidity. They are used by researchers to evaluate environmental changes, and their metabolic indicators are used as functional factors for the health of river ecosystems [11]. Epilithic biofilms can generally respond to changes in the surrounding environment and reflect changes in the benthic algal communities’ composition through changes in their biomass.

Among the numerous studies of the South-to-North Water Diversion Project, most focused on the discussion of the space–time evolution of water quality indicators and ecological health assessment [12], but fewer have focused on aquatic organisms. Some scholars investigated the space–time evolution of algae in the middle route of the South-to-North Water Diversion Project [13,14], and some scholars discussed the impact of hydraulic structures on algae [15]. However, there has been little research on the removal of hydrodynamic algae, especially benthic algae. The effect of hydrodynamic conditions on algal growth is that a low-flow velocity promotes algal aggregation and attachment [16], whereas medium-intensity disturbances increase nutrient uptake by algal cells and promote metabolism [17]. In contrast, high-intensity disturbances inhibit algal growth, nutrient absorption, and cell metabolism [18]. The hydrodynamic conditions affect changes in algal cell proliferation, morphology, metabolic activity, and nutrient uptake. Additionally, they affect the algal community structure, population succession, biomass distribution, and critical velocity. Studying the influence of hydrodynamic conditions on algal physiology and ecology can play a key role in controlling algal blooms.

Various organisms constitute aquatic ecosystems, and they play an important role in river ecosystems [19,20]. Benthic algal communities in aquatic ecosystems are highly sensitive to changes in the living environment [21]. Changes in the benthic algal community reflect changes in the ecosystem structure and function. Therefore, we hypothesized that flow velocity changes could be widely applied in open channels to reduce benthic algal biomass and alter community structure. In this study, we diverted water into the open channel simulating the SNP by monitoring environmental factors based on benthic algal cell identification and defining the historical succession of benthic algal communities. The impact of water diversion on environmental factors and benthic algal community structure in the SNP open channel was explored. We aimed to (1) reveal the main influencing factors of the open channel river ecosystem, (2) reduce the risk of benthic algae outbreaks by regulating the flow, and (3) prove the feasibility of regulating the flow to ensure the safety of drinking water. This will provide a good solution for benthic algal blooms caused by human activity.

## 2. Methods

### 2.1. Experimental Setup and Design

We used a univariate study design, with traffic as the only variable. A 35-day artificial river mesoscale simulation experiment was conducted downstream of a small power station in the Yongding River Diversion Canal in Beijing, China. To study the response of algal communities to three different flow conditions, we built a straight channel with a length of approximately 40 m (made of cement mortar and masonry), a width of 0.6 m, and a depth of 0.5 m (Figure 1). Additionally, we also made thick water pipes with a length of >20 m and an inner diameter of 0.1 m as water delivery containers. The experimental water was taken from under the Yongding River Barrage by eight submersible pumps with a flow regime of 10 m^3^/h and two sets of 25 m^3^/h linked to the water delivery container for redistribution, facilitating the control of the total diversion flow. The water was redistributed and tapped on the four channels by the water pipe for use. We also installed a regulating valve at the inlet of each channel to regulate the water level stability and the flow regime step change by modifying the valve opening to have four different flow conditions. Four sets of 7 × 9 cm marble tiles provided by the supplier were placed symmetrically in the middle of the channel section. Before starting the experiment, the channel was run with water for 2 weeks to allow benthic algae to grow, the channel was exposed to sunlight with a wide view, and the channel bottom did not contain any mineral or organic sediments. In May 2022, we measured the lowest velocity of benthic algae growth area at 0.086 m/s, so we chose 0.086 m/s as the basic velocity, and the maximum velocity was five times that of the base velocity. During the experiment, we kept the base velocity unchanged.

### 2.2. Sample Extraction and Measurement

#### 2.2.1. Water Sample Collection

As benthic algae integrate nutrients, changes in river nutrient conditions affect the benthic algal community composition. Therefore, our monitoring frequency of water quality was higher than the sampling frequency of benthic algae to ensure there was no large difference in environmental changes. We collected water samples from the benthic algae-attached matrix in each river section in 150 mL polyethylene bottles. Subsequently, the water samples were transported to the laboratory for filtration using Whatman GF/F filters (0.22 µm pore size). Next, the water was stored at <4 °C (not frozen) and tested within 24 h.

#### 2.2.2. Benthic Algae Sample Collection

Four groups of artificial substrates were placed side-by-side in the central area of each experimental interval and marked. We collected several equal areas of benthic algae and mixed them into one sample for measurement to reduce experimental errors. A polyvinyl chloride tube with an outer diameter of approximately 3.7 cm was placed in each group of artificial substrates. The benthic algae in the area were scraped with a toothbrush and rinsed with distilled water into a wide-mouth plastic bottle as a quantitative sample of benthic algae. The remaining benthic algae were scraped into another wide-mouth plastic bottle, transported back to the laboratory, and added to Lugol’s solution and 4% formaldehyde solution for fixed preservation as qualitative samples of benthic algae.

#### 2.2.3. Physical and Chemical Analysis

Water temperature, pH, dissolved oxygen (DO mg/L), and other parameters were measured on-site using a calibrated portable multiparameter water quality analyzer (Multi 3630 IDS, WTW, a subsidiary of Xylem Inc., Weilheim, Germany). We took an appropriate number of water samples to the lab to measure total phosphorus (TP mg/L), total nitrogen (TN mg/L), nitrate nitrogen (NO_3_^−^ mg/L), nitrite nitrogen (NO_2_^−^ mg/L), and ammonia nitrogen (NH₄^+^ mg/L) contents, following the standard method recognized by USEPA, using a UV spectrophotometer (DR6000, HACH, Loveland, CO, USA). The flow velocity was monitored on-site using a portable flow velocity meter (LS300-A, Nanjing Xiangruide Electric Technology Co., Ltd., Nanjing, China).

#### 2.2.4. Benthic Algae

Benthic algae species were identified according to the European standard method EN 14407 [22]. The algal species were identified and counted under a 1000× optical microscope. The number of benthic algal cells observed on each coverslip was >1000. The identification and naming of benthic algae were based primarily on “Chinese freshwater algae” published by the Chinese Academy of Sciences.

Benthic algae were quantified using spectrophotometry. This involved filtration through a 0.45 μm glass fiber membrane, grounding to disrupt algal cells, extraction with acetone for 2–24 h, and the absorbance measurement of the centrifuged supernatant at 750, 664, 647, and 630 nm. Additionally, we ensured that the whole process was processed in a shaded environment.

### 2.3. Statistical Analysis of Data

We performed a one-way ANOVA comparison of benthic algal biomass measurements (Chl-a and cell density) to assess differences in flow velocity treatments. The absolute growth rate during the experiment was inferred from the rate of change in the biomass sampling process. Furthermore, the dominant species and richness of benthic algal communities were quantitatively assessed using the benthic algal dominance index and the Simpson diversity index. We performed correlation analysis using Spearman’s correlation analysis for TN, TP, NH_4_^+^, NO_3_^−^, and NO_2_^−^. Additionally, we performed decision curve analysis on species samples and determined whether to use canonical correlation analysis or redundancy analysis (RDA) according to the size of the first coordinate axis of the gradient length. Lastly, we determined the primary environmental factors driving the change in benthic algae community structure. The results were statistically analyzed and plotted using Origin 2022 Pro and Canoco5.0 software.

## 3. Results

### 3.1. Evidence for Regime Alterations from Benthic Algal Biomass Levels

The temporal evolution of Chl-a (μg/cm^2^) in the four flow sections of the experiment is shown (Figure 2a); the effect of different flow velocities on Chl-a among benthic algae on the seventh day was not significantly different (0.05 < *p* < 0.7). However, the effect was significant on the 14th day (ANOVA, *p* < 0.05), the 21st day (ANOVA, *p* < 0.05), and the 28th day (ANOVA, *p* < 0.05). At the end of the experiment, the biomass of the F4 section (0.418 m/s) was much lower than that of the F1 (0.086 m/s) and F2 (0.125 m/s) sections. Throughout the study period (Figure 2b), biomass levels were sensitive to water flow velocity. The median and mean values of Chl-a first increased and then decreased. The statistical value of the highest-velocity section (F4) was the lowest. Additionally, the overall benthic algal biomass of F4 was much lower than that of F1 after the river ecosystem reached specific flow values (F3 and F4). This shows that the overall benthic algal biomass is significantly reduced with increased flow.

Regarding community structure, there were significant differences in the flow velocity and the number of species. Additionally, the colonization of surface rock biofilms in the F3 and F4 canals was significantly delayed. In the first week of the experiment, the high-flow velocity group had only half of the community species in the low-flow velocity group, and the biomass level was only 1.87 μg/cm^2^. This shows that in the initial colonization stage, the highest water flow resistance slowed down the deposition of the epiphytes of benthic algal spores, resulting in the colonization effect under medium- and low-flow velocity, which was better than that under the high-flow velocity condition. From the second week of the experiment, the medium- to high-flow velocity accelerated the rate of metabolic waste diffusion and material exchange between the water body and algal cells, increasing the growth rate of the attached algae.

### 3.2. Environmental Metrics Responses to Gradient Flow

The environmental factors and Spearman’s correlation analysis of the four sampling points before and after the experiment are shown in Table 1 and Figure 3. During the experiment, we found that the changes in environmental factors are mainly reflected in the changes in the form of nitrogen elements in the water body, among which organic nitrogen content fluctuates greatly; this may be due to the change in the flow velocity, which changes the microbial community in the water, thus affecting the nitrogen cycling in the water. At the same time, the available nitrogen elements of algae are reduced (such as NH4^+^), which indirectly proves that the change in the flow velocity reduces the biomass of benthic algae; TP and temperature remained stable with the increased gradient flow velocity. Furthermore, with an increase in flow gradient, the two environmental factors, pH and DO, showed a significant positive linear correlation (Spearman, *p* < 0.05); however, TN and NH_4_^+^ were significantly negatively correlated (Spearman, *p* < 0.05). This signifies that pH and DO increased linearly with an increase in flow velocity; however, TN and NH_4_^+^ decreased under the same condition. Additionally, the change rule of environmental indicators with biomass was opposite to that of the flow velocity. The change in flow velocity significantly reduced the biomass of benthic algae, and TP also showed a negative correlation. Overall, the results of the environmental factors show that with the change in flow velocity gradient, various indicators in the simulated channel showed a relatively positive response, with improvement in the water quality. Changes in water eutrophication did not significantly affect the diversity of benthic algae.

### 3.3. Relationship between Flow Gradient and Taxonomic Composition of Benthic Algae

A total of 48 species of benthic algae from 4 phyla, 5 classes, 11 orders, and 29 genera were identified, of which 30 species were identified from the diatom phylum, accounting for 62.5%, and only 1 species of dinoflagellates, accounting for 2.1%. There were 6 species of cyanobacteria, accounting for 12.5%, and 11 species of Chlorophyta, accounting for 22.9%. Among the diatoms, nine species of *Navicula* were identified, among which *Navicula minuscula*, *Navicula* sp., and *Navicula minima* had the highest frequency.

The experimental water body was a diatom-green algae-cyanobacteria water body. The α diversity index calculated for the benthic algae in each channel section is listed in Table 2. Specifically, in the distribution of Chl-a, the increase in the gradient flow velocity first increased its content, which then decreased rapidly at the flow velocity channel section of 0.418 m/s. The changes in the biodiversity index, including the number of community species, showed the same trend. The biodiversity index first increased and then decreased, and the peak appeared at the F3 sample point. This means that when the flow velocity increased to 0.211 m/s, the richness, diversity, and species number of the benthic algae community reached a maximum. However, the flow velocity continued to increase the biodiversity of the F4 sample point (v > 0.30 m/s), and the community richness and diversity index decreased to a certain extent. Specifically, the Shannon biodiversity index increased from 0.17 to 1.95, and the Simpson index increased from 0.05 to 0.74 and then decreased to 0.38 and 0.14 at F4. The Pielou and chao1 indices of community richness increased from 0.06 and 20.00 to 0.61 and 30.00, respectively, and then decreased to 0.13 and 20.50.

During the first week of the experiment, the colonization pattern of benthic algae in the low-velocity group on the left was more regular, showing a uniform and the same spatial distribution on each artificial marble substrate, indicating a uniform adhesion and fixation effect. In contrast, the distribution of benthic algae in the artificial marble matrix in the high-velocity group was loose, showing a central patch on the front vertical upstream surface and a random distribution along the horizontal plane. Initially, benthic algae spread approximately vertical to the surface of the attached substrate along the initial central patch of the vertical plane facing the water, and then slowly diffused throughout the attached matrix with the timetable, consistent with the flow conditions around the cuboid. Additionally, top-view video images and biomass measurements revealed that biomass accumulation decreased with increasing water flow velocity during the colonization stage.

The Sankey diagram (Figure 4) shows more directly the changes in species relationships. The different streamlines indicate obvious differences in the algae canal section. Before and after the experiment, the high-velocity environment promoted an increase in the relative abundance of most benthic algae with strong adhesion, such as *Cocconeis placentula*; in contrast, the low-velocity environment gathered most planktonic or weakly attached algae (e.g., *Gloeocapsa* sp.).

As the experiment proceeded, the observed epiphytic pattern was that epilithic biofilms formed at lower flow velocity were thicker and had higher cell density per unit area than biofilms formed at higher flow velocity, consistent with numerous other microbial studies [23,24]. There were obvious differences before and after the experiments in the shape characteristics of algal filaments: the thick biofilm mat of the low-velocity group was located at the front end of the flow velocity gradient, and filamentous algae, which hardly appeared, showed a thick mud-like biofilm shape in the medium-flow velocity section located downstream. Occasionally, algal filaments agglomerated and gradually spread to the entire plane, extending to a maximum length of more than one plane (approximately 12 cm); at the end of the entire channel, where the flow velocity was the highest, the algal filaments were elongated to a length of at least 30 cm and were preferentially attached to the rough substrate. Of all benthic algae samples, *Gloeocapsa* sp. showed the highest frequency. However, regarding biomass, *Cocconeis placentula* had the most biomass in the F1 section, and the biomass of *Cladophora* sp. was the largest in the middle- and high-velocity channel samples. Furthermore, the Venn diagram (Appendix A) shows species-level differences and the community structure at the initial stage and the end of the experiment, indicating differences in the number of species between each channel section. A total of 48 species of benthic algae gathered in four different gradient flow velocity environments, and the F1, F2, F3, and F4 canal sections, had 3, 9, 3, and 3 unique species, respectively. These sampling points shared 15 species (31.25% of the total).

Before the flow velocity reached 0.418 m/s, the relative abundance of benthic algae cultured at medium and low velocity was comparable and showed a certain trend. After a large-scale algal bloom, the abundance of dominant species spread from a high-disturbance environment to a low-disturbance environment. None of the benthic algae metrics we focused on had a linear response to flow velocity throughout the experiment. When we looked at the algae changes in those major proportions, some interesting information emerged. At the beginning of the experiment, most of the non-dominant species, such as *Cocconeis placentula, Cyclotella* sp., and *Synedra* sp. in the diatom phyla in the whole sample showed a trend of increasing relative cell density along the gradient to the flow velocity. However, at the fourth sampling point, the channel section with flow velocity >0.4 m/s, the relative abundance decreased to a very low level. *Gloeocapsa* sp. species with a very high frequency showed the opposite change: the relative cell density decreased with the change in flow velocity gradient (the variation range was 0.40–4.04 × 10^6^ cells/cm^2^, accounting for 72.24–98.39% of the total number of cells). Moreover, in the channel section with a flow velocity of 0.211 m/s, the relative cell density increased to 2.53 × 10^6^ cells/cm^2^. Samples at the end of the experiment showed the same results as during the initial stage but with a higher reduction in Myxococcus with the flow velocity gradient. Under the action of water flow, Myxococcus only maintained a relative cell density of 47.16% at a flow velocity of 0.211 m/s; however, it did not change the dominant species of the community.

To better express the community response of benthic algae from different flow velocity gradients, we drew a cluster heat map of 48 benthic algae species to analyze the changes in community structure in detail, as shown in Figure 5. At the F1 site, *Cocconeis placentula* had the highest relative abundance, and *Pinnularia* sp., *Tetraëdro tumidulum*, and *Cymbella perpusilla* were clustered together, and did not often settle in the water body. Occasionally, they dispersed on the water surface and were often enriched in low-velocity waters. However, when the flow velocity increased (F4), the relative abundance of *Cladophora* sp. was the highest, and the transformation was attributed to the living habits of the algal species. This species belongs to higher algae, and has the same chlorophyll structure as aquatic plants and is fond of epiphytic animals and water environments [25]. Furthermore, it has strong adaptability to a low-CO_2_ environment, with the ability to obtain a carbon source directly from HCO_3_^−^. Species such as *Fragilaria* sp., *Nitzschia fonticola,* and *Gomphonema olivaceum* were grouped in a taxon classified as a species found in human-influenced rivers [26].

We performed RDA to identify major environmental variables associated with changes in the benthic algal community structure (Figure 6). The first two axes represent 27.43% of RDA1 and 26.13% of RDA2 for benthic algal community structure changes, indicating that selected environmental factors drove differential changes in the benthic algal community structure. Of all the basic indicators studied, flow velocity had the most significant impact on benthic algae and can explain most of its community structure changes. In the RDA score chart, it can be seen that the distribution of the four groups of community sample groups is widely spread, which reflects the great difference between sample communities in each section of the open channel with a velocity gradient. The flow velocity arrow and the origin are the longest, indicating that the correlation between flow velocity and community distribution in species distribution was the largest and negatively correlated, explaining 25.4% of the variance components of the benthic algae community changes (*p* = 0.01). This further demonstrates that an increased flow velocity reduces the benthic algae levels. In conclusion, flow velocity greatly influences the diversity of the benthic algae community and is the main indicator affecting the change in benthic algae in the channel.

## 4. Discussion

Benthic algal biomass levels in river ecosystems provide key evidence for assessing ecosystem stability [27,28,29]. Conventional ecological theory suggests that environmental resources modulate benthic algal community composition [30]. Consequently, many studies have investigated the impact of environmental habitats on benthic algal communities, and many ecological assessment methods have been developed [31,32]. However, there is still a lack of a clear understanding at the physical level, such as in hydrodynamics, especially since the research results caused by local differences are not universal [33]. Few studies have documented the response of benthic algal composition to ecologically significant hydrological properties. Specifically, only a few studies have analyzed the relationship between algal diatom assemblages and changes in river flow conditions [34,35]. In this paper, we discussed the response relationship of gradient flow velocity to the benthic algae community and the response of environmental factors to the water flow by simulating the changes in water volume in artificial streams and the benthic algal community structure. Through biological-level analysis—microscopic examination and chlorophyll spectrophotometry—the main categories of benthic algae were found to include prokaryotic cyanobacteria, lipid-rich diatoms, and microscopic and macroscopic green algae, which are common in natural rivers and lakes [36,37,38].

The changes in the biomass and composition of benthic algae demonstrate that different flow velocity gradients have a certain correlation between the structure of epilithic biofilms and the composition of algae. Epilithic biofilm structure and benthic algal composition can be expressed as local hydrodynamic growth relationships [39]. Spearman correlation analysis showed a negative correlation between flow velocity and benthic algal biomass. Compared with the low-flow velocity environment (about 0.1 m/s), the high-flow velocity environment increased the growth rate of algae to a certain extent. However, it shortened the algae growth cycle, and the biomass peak lasted only about a week.

This study focused on the effects of water flow conditions on the growth and composition of benthic algal communities. The flow velocity and suspended matter may play a non-negligible role in regulating the ecological relationship of epiphytes. In natural rivers, there are flood pulses and high-intensity flow disturbances, which generate large shear stress on the boundary layer of algal mats [40,41], and affect the wear and turbidity of suspended matter [42,43]. Inhibiting the growth, nutrient absorption, and cell metabolism of benthic algae destroys the attachment conditions of algal mats, thus controlling the excessive proliferation of benthic algae. However, researchers have often focused on ecological thresholds [44,45], ignoring local differences and differences in primitive epiphytic conditions. Epiphytic benthic algae generally have significantly different anti-flushing abilities at different flow velocities [39]. Epiphytic benthic algae in a high-velocity environment have stronger adsorption than those in a low-velocity environment, are more resistant to denudation caused by liquid shear force, and have less biomass loss. However, high water flow slows the colonization of benthic algae and reduces the peak biomass. In this study, the high-velocity channel section (v = 0.418 m/s) significantly reduced the level of benthic algae Chl-a. In this study, the open channel lacked fine sediment and did not experience the light attenuation and wear effects caused by fine sediment particles suspended and moved due to flow disturbance.

The alpha diversity index reflects the microbial communities’ species richness, and diversity is commonly used to characterize algal communities [46,47]. This study explored the relationships between water flow, benthic algal community structure, and biodiversity. From α diversity index analysis, each diversity index showed the same trend of change with the flow velocity gradient, which peaked at the 0.211 m/s flow velocity channel section and then decreased at the 0.418 m/s flow velocity, showing a rising and falling trend. Moreover, we found that the measurement results of the biomass proxy indicator Chl-a showed that high-flow velocity significantly reduced biomass. However, this shows that high-velocity water flow increases the biomass accumulation rate to a certain extent. Water flow under high-velocity scouring can increase the growth of benthic algae [48,49]. In this study, the peak time of terrestrial algae biomass in the high-velocity canal section was earlier than that in the low-velocity canal section. This is because high-velocity water flow accelerates material exchange between algae and the water body [50,51]. However, in actual research, high- velocity environmental water flow sometimes cannot effectively remove benthic algae; however, it will lead to an increase in filamentous taxa.

A poor-quality large flow velocity environment can easily take away planktonic algae. Furthermore, a non-long-term stable flow environment is insufficient to create a benthic algal community. However, the persistent low flow may result in poor-quality aquatic communities and should be avoided in practical river management.

Under normal circumstances, the benthic algal community in still water exhibits resource specificity. Species that can obtain sufficient nutrient resources have a competitive inhibitory relationship with other species in the environment; however, some species are not sensitive to water quality requirements. *Cocconeis placentula*, common in this study, occurs in polluted and unpolluted rivers. Changes in the river environment (e.g., channel width, flow velocity, depth) and physicochemical characteristics (e.g., DO, water temperature) affect the transformation of benthic algal life forms and communities [52], shifting from taxa with planktonic or weak attachment in low-velocity waters (F1) to advanced taxa with strong attachment in high-velocity waters (F3, F4).

In our research, we found morphological differences in benthic algae during the third week of the experiment. Filamentous green algae appeared in the highest flow velocity environment and continued to spread upstream. This phenomenon indicates that the change in flow velocity with time caused the establishment of benthic algae for the succession of large filamentous algae. This is a unique colonization pattern of algae. In new environments, substrate conditions may not be suitable for developing filamentous green algae [53]. Anti-scour algae, such as diatoms, are preferentially attached as early pioneers [54,55]. With the stacking and development of the community, the conditions of the benthic substrate gradually integrated, and filamentous green algae began to form. The filamentous green algae may survive in high-velocity environments. The metabolic rate of the cell membrane in a high-velocity environment was higher than that in a low-velocity environment, resulting in a higher benthic algae growth rate in high-velocity environments than that in low-flow-velocity environments. In the morphologically high-velocity environment, the algal filaments were longer, and the branches were denser. This type of filamentous green algae with a branched structure is intertwined and is not easily eroded; however, it may be subject to the action of hydrodynamics. Through algal cell propagation, algal filaments are continuously elongated and sometimes reach a length of approximately 30 cm. These benthic algae species can easily attach to the rough attachment surface with pores. The artificial concrete substrate is most likely to be colonized and attached by filamentous green algae, often accompanied by an outbreak of points and surfaces. For these troubling and difficult-to-handle benthic algae, we considered reducing their spore/gamete attachment ability by disposing of the artificial coating in follow-up research to completely solve the problem of algal blooms in open channels.

Our research does not cover the whole-year benthonic algae monitoring scale; however, this does not affect our conclusion that the benthonic algae biological level decreases within a certain flow velocity range. At the same time, our research lacks treatment with a higher flow velocity, which is caused by inadequate equipment conditions resulting from backwater due to the insufficient slope of the open channel. In the next step, we will supplement the limitations of this study by monitoring benthic algae culture for a long time and regulating the open channel with a high-flow velocity. Therefore, water disturbance can regulate the aggregation properties of algal cell populations. Relative to the highly disturbed water environment, the river flow is reduced, and the flow velocity is slowed down, providing the conditions for the reattachment of the originally suspended benthic algae and increasing the risk of a benthic algae bloom attached to the bottom layer. The regulation of flow velocity is the most convenient measure to manage the river, especially in long-distance water delivery projects. The benthic algal epiphytic conditions are changed by regulating the release flow, thereby reducing the risk of benthic algal over-proliferation.

## 5. Conclusions

We found that the environmental factors pH, TP, and DO all showed a positive response to the flow velocity, whereas TN, NO_3_^−^, and NH_4_^+^ did not. This is because the flow velocity is negatively correlated with benthic algae biomass. Furthermore, the mean and median benthic algal biomass gradually decreased in the simulated river ecosystem when the regulated flow velocity was >0.418 m/s. This was significantly lower than the overall biomass at the initial flow velocity (0.086 m/s), and approximately 39.88% of the biomass was reduced. During the 35-day flow regulation period, the community structure converged towards monodominant algae, and macroscopic and planktonic algal cells were transformed into filamentous algae. Furthermore, the number of unique algae species in the open channel decreased, and the indicators of biodiversity and richness decreased to varying degrees. This means that increasing the flow velocity will promote the transformation of algae to filamentous green algae and inhibit the proliferation of algae to a certain extent. Moreover, in RDA, the flow velocity is the main environmental variable causing changes in the benthic algal community structure and is the main indicator affecting the change in benthic algae in the channel. By regulating flow velocity, the occurrence of algal blooms in river ecosystems can be suppressed. This provides a theoretical basis for managing flow velocity to slow down the excessive proliferation of benthic algae in rivers. Our research lacks treatments with a higher flow velocity and longer monitoring period, so the conclusion cannot provide a guiding value for autumn and winter seasons.

## Figures and Tables

**Figure 1 ijerph-20-03564-f001:**
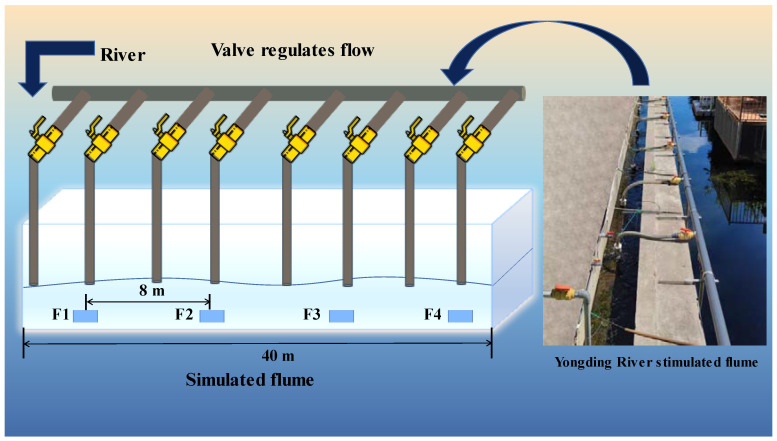
Schematic diagram of the experimental water tank.

**Figure 2 ijerph-20-03564-f002:**
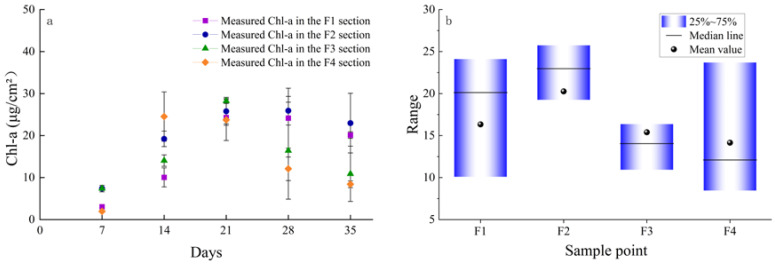
Evolution of the Chl-a ± SE (μg/cm^2^) in the four flow velocity gradient sections (F1, F2, F3, and F4) on different days (**a**) and total average (**b**) after the experimental operation.

**Figure 3 ijerph-20-03564-f003:**
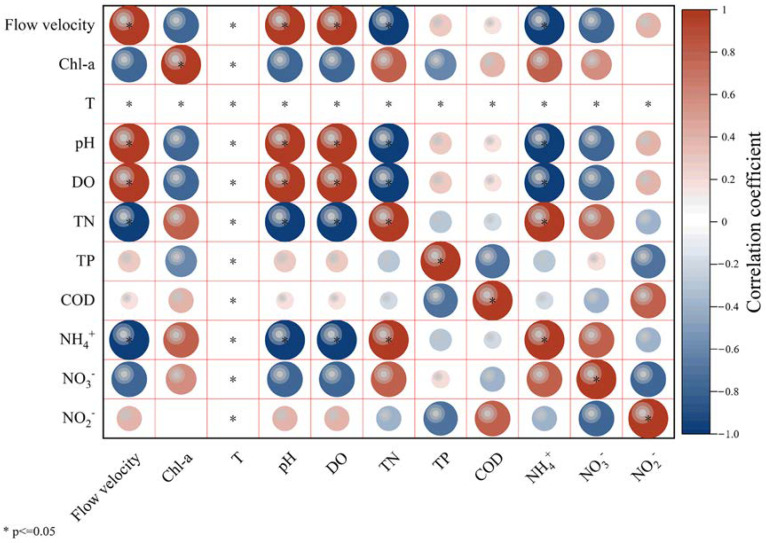
Spearman correlation heat map of environmental factors and benthic algae biomass (* indicates two indicators are significantly correlated, *p* ≤ 0.05. The larger the circle, the stronger the correlation between the two indicators).

**Figure 4 ijerph-20-03564-f004:**
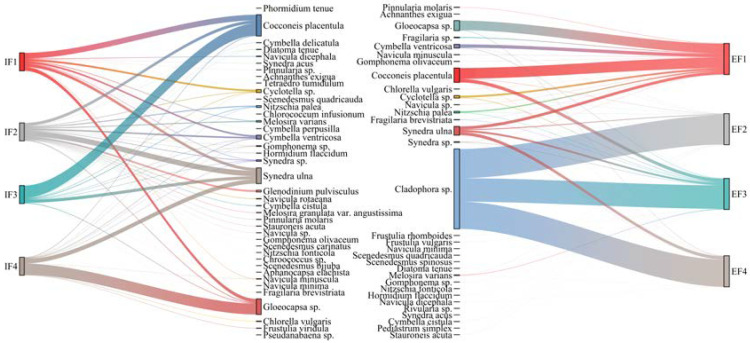
Sankey diagram of the abundance of benthic algae in four flow velocity gradient sections before and after the experiment.

**Figure 5 ijerph-20-03564-f005:**
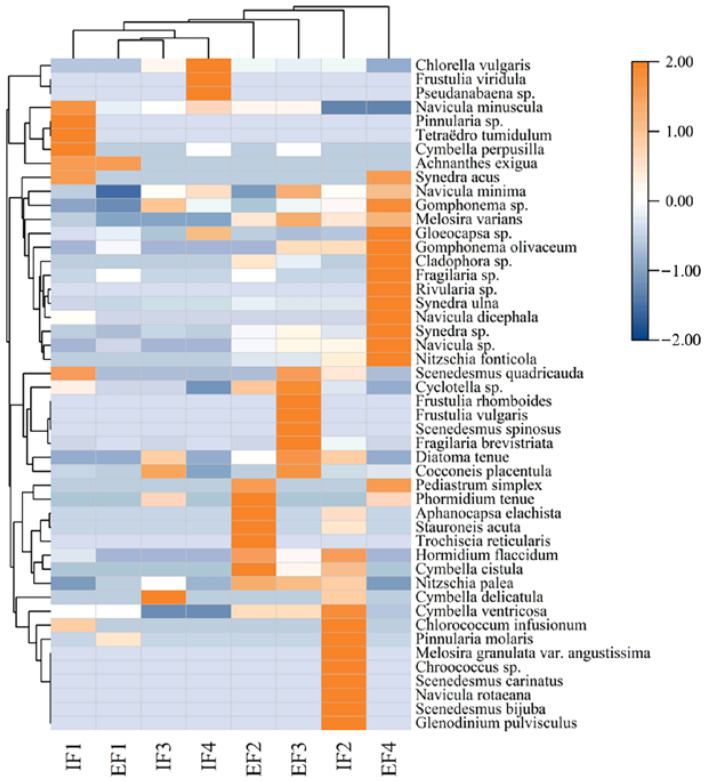
Hierarchical clustering heat map of benthic algae abundance level treated with four flow velocities (Note: IFi represents the sample at the initial stage of the experiment, and EFi represents the sample at the end of the experiment. The higher the relative abundance of the species, the brighter the color).

**Figure 6 ijerph-20-03564-f006:**
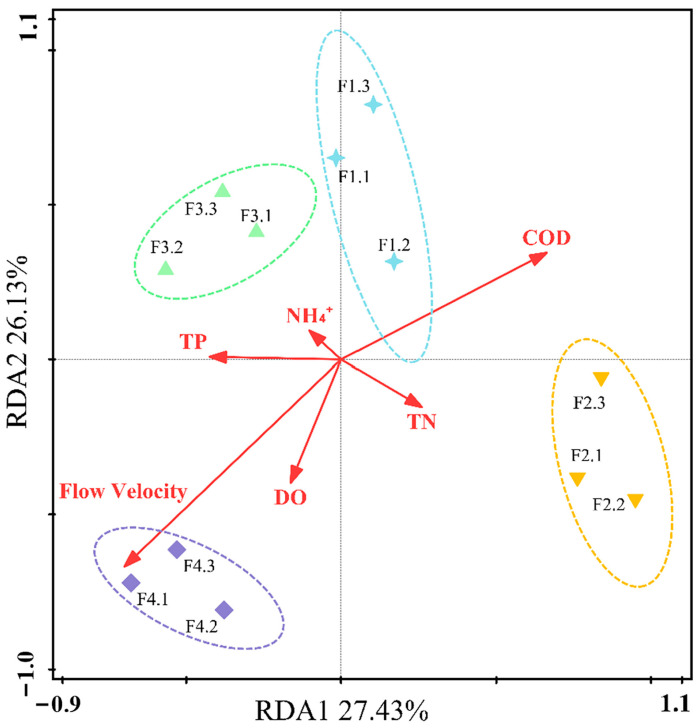
Biplot diagram of redundancy analysis (RDA) of benthic algae biomass, abundance, and environmental variables. Red vectors: environmental factors, color ellipses: grouping ellipses of F1, F2, F3, and F4 repeated samples.

**Table 1 ijerph-20-03564-t001:** Statistics of water environment indicators under gradient flow velocity before and after the experiment.

Environmental Factors	The Initial Stage of the Experiment				The End of the Experiment			
	F1	F2	F3	F4	F1	F2	F3	F4
Flow velocity (m/s)	0.086	0.125	0.211	0.418	0.086	0.125	0.211	0.418
T (°C)	15.84	15.84	15.84	15.84	18.40	18.40	18.40	18.40
pH	7.520	7.533	7.545	7.564	7.582	7.569	7.564	7.570
DO (mg/L)	10.42	10.46	10.49	10.35	9.92	9.99	10.05	10.10
TN (mg/L)	14.34	13.25	12.85	12.77	5.05	6.20	4.60	5.25
TP (mg/L)	0.11	0.10	0.10	0.11	0.09	0.08	0.10	0.11
COD (mg/L)	27	34	33	30	31	33	31	26
NH_4_^+^ (mg/L)	0.14	0.08	0.07	0.04	0.13	0.14	0.21	0.15
NO_3_^−^ (mg/L)	6.60	6.45	5.25	6.17	4.30	4.10	4.55	4.45
NO_2_^−^ (mg/L)	0.048	0.076	0.083	0.052	0.013	0.013	0.013	0.014

Abbreviations: DO, Dissolved Oxygen; TN, Total Nitrogen; TP, Total Phosphorus; COD, Chemical Oxygen Demand; NH₄⁺, Ammonia Nitrogen; NO_3_^−^, Nitrate Nitrogen; NO_2_^−^, Nitrite Nitrogen.

**Table 2 ijerph-20-03564-t002:** Comparison of benthic algae community characteristics at a different flow velocity.

Sample	Abundance	Shannon	Simpson	Pielou	Chao1
F1	15	0.17	0.05	0.06	20.00
F2	24	1.06	0.38	0.33	25.20
F3	25	1.95	0.74	0.61	30.00
F4	19	0.38	0.14	0.13	20.50

## Data Availability

The dataset analyzed during this study is available from the corresponding author upon reasonable request.

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
