# Peer review of "Reducing the Risk of Benthic Algae Outbreaks by Regulating the Flow Velocity in a Simulated South–North Water Diversion Open Channel"

_ijerph, 2023, doi:10.3390/ijerph20043564_

Round 1

Reviewer 1 Report

Dear authors,

your study is well-performed and I can suggest it for publication. I only have one question / suggestion for discussion: For implementation to practise- how would it be possible to enhance flow if the slow in the channel is low to limited slope? Would you pump the water? Another possibility to avoid algal growth is shading (closed channels, shading walls).

Best regards

Your reviewer

Reviewer 2 Report

The manuscript entitled “Reducing the risk of benthic algae outbreaks by regulating the
flow velocity in a simulated South-North Water Diversion open channel” has been reviewed. The subject is interesting and falls within the scopes of IJERPH. In my opinion the manuscript could be accepted considering the following comments.

Abstract

Please avoid using abbreviation in the abstract or define it. Also, the font type and size should be uniform throughout the abstract.

Methods

Page 3: “The channel was run with water for 2 weeks to allow benthic algae to grow, the channel was exposed to sun-light with a wide view,” How one can make sure that 2 weeks are enough? What happens if the treatment lasted more time? How many hours were the sun shining in each day?

Page 3: “The base flow was 0.086 m/s” please revise as “The base flow velocity was 0.086 m/s” and do the same changes for the rest of the text. Also, why you used these values for flow velocity? You should provide some proofs of the similarity between the experimental and real conditions.

Results

Page 6: why nitrogen elements are more affected? Please discuss with more details.

Page 6 Figure 3: please add the right axis title. Also, what does the size of the  red and blue circles in this figure show? Please explain in the text.

Page 9: Figure 5: please add the name of the studied parameter and its unit in the legend.

Page 12: please replace “high-speed” with “high-velocity

Conclusions:

Please add the limitation(s) of the present study at the end of the conclusions.

Reviewer 3 Report

This paper is interesting and well written. It could provide a good solution for benthic algal blooms caused by human activity. Thus, minor revision is recommended.

1.      The abstract should be stand alone. It is suggested to further elaborate F3 and F4 so that readers can understand the main content of this work after reading the abstract.

2.      It is suggested to add several relevant studies on SNP in the introduction.

3.      I It is suggested to emphasize the novelty of this work in the introduction.
